# Censoring Representations with Multiple-Adversaries over Random Subspaces

**Yusuke Iwasawa, Kotaro Nakayama & Yusuke Iwasawa**
Department of Engineering, The University of Tokyo
{iwasawa,nakayama,matsuo}@weblab.t.u-tokyo.ac.jp

## Abstract

Adversarial feature learning has been successfully applied to censor the representations of neural networks; for example, AFL could help to learn anonymized representations to avoid privacy issues by constraining the representations with adversarial gradients that confuse the external discriminators that try to discern and extract sensitive information from the activations. In this paper, we propose the ensemble approach for the design of the discriminator based on the intuition that the discriminator need to be robust to the success of the AFL. The empirical validations on three user-anonymization tasks show that our proposed method achieves state-of-the-art performances in all three datasets without significantly harming the utility of data. We also provide initial theoretical results about the generalization error of the adversarial gradients, which suggest that the accuracy of the discriminator is not a deterministic factor for the design of the discriminator.

## 1 Introduction

When we apply deep neural networks or more general machine learning techniques to real-world data, one of the key challenges is how to systematically incorporate the desired constraints (such as privacy (Edwards & Storkey, 2016; Iwasawa et al., 2017)) or fairness constrains(Zemel et al., 2013; Louizos et al., 2016)) into the learned representations in a controllable manner. One of the recently proposed approaches for censoring representation is *adversarial feature learning* (AFL) (Edwards & Storkey, 2016; Iwasawa et al., 2017; Xie et al., 2017), which considers an adversarial classifier that attempts to discern sensitive variables from the data representations in a DNN and simultaneously trains the DNN to deceive the classifier. Formally, AFL constrains the output of an encoder $z = E(x)$ to reduce the approximated likelihood about a sensitive variable $s$ while maintaining the classification performance about the label $y$:

$$\min_{E,M} \max_{D} \mathbb{E}[-\log q_M(y|z = E(x)) + \lambda \log q_D(s|z = E(x))],$$

where $\lambda$ is a weighting parameter, and $q_M$ and $q_D$ are conditional distributions parameterized by the label classifier $M$ and the adversarial classifier $D$ respectively. By alternatively or jointly (using gradient reversal layer Ganin & Lempitsky (2015)) training the adversary and DNN in such a manner, AFL ensures that there is little information about the sensitive variables in the representations.

In this paper, we seek to answer the following questions: what property should the adversary possess to improve the performance of AFL? Although some previous studies report superior performance of the AFL, the success of the AFL depends on the choice of the adversarial classifier. For example, if we use logistic regression as the adversarial classifier, AFL cannot remove any non-linear dependency by their notion. It is also possible that deceiving some classifiers might be relatively easy, resulting in poor performance improvements. This paper makes the following contributions to the design of discriminator. (1) This paper proposes a novel design for the adversary, *multiple-adversaries over random subspace* (MARS), based on the intuition that the discriminator need to be hard to deceive. (2) This paper provides theoretical analysis about the generalization error on the effect of the adversarial gradients, suggesting that the accuracy of the discriminator is not a deterministic factor for the design of the discriminator. (3) Empirical validations show that proposed method could learn better-anonymized representation without performance drop on the label-classification.

## 2 ROBUSTNESS OF DISCRIMINATOR IN AFL

### 2.1 MULTIPLE ADVERSARIES OVER RANDOM SUBSPACES

The proposed method, *multiple-adversaries over random subspaces (MARS)*, considers multiple adversaries where each adversary is in charge of different subsets of features. The MARS is motivated by the assumption that the adversary should not be vulnerable and an ensemble of diverse classifiers make the adversary less vulnerable, resulting in the improved performance of AFL.

Suppose that $n$ is the number of dimensions of representation $R \in \mathbb{R}^n$, and $K$ is the number of adversaries. In MARS, each adversary $D_k$ is trained to predict $S$ over a randomly selected subset of features $R^k$, whose dimension is $m_k$ and $m_k < n$. Each adversary is trained to maximize the expected log-likelihood function:

$$\max_{\theta_{D_k}} \mathbb{E}[\log q_{D_k}(s|h_k = Sub_k(E(x)))], \tag{1}$$

where $Sub_k$ is a function that return the subset of $R$, which is fixed before the training. Precisely, each $Sub_k$ determine whether to remove the $i$-th dimension of $R$ with the probability of $\alpha$. The $\theta_{D_k}$ is the parameter of $D_k$. The encoder is then trained with: $\min_{\theta_E, \theta_M} \mathbb{E}[-\log q_M(y|h) + \lambda \log \frac{1}{K} \sum_k^K q_{D_k}(s|h_k)]$, where $\theta_E$ and $\theta_M$ are the parameter of $E$ and $M$ respectively.

### 2.2 GENERALIZATION ERROR OF THE EFFECT OF ADVERSARIAL GRADIENTS

Suppose $\nabla_{adv} D$ is the adversarial gradients of the current discriminator $D$. After one iteration of the above optimization process, we can measure the extent to which the adversarial gradients decrease the predictability of $s$: $V_D(\nabla_{adv} D) = \log q_D(s|E(x); \mathcal{D}) - \log q_D(s|E'(x); \mathcal{D})$, where $E$ and $E'$ is the pre/post updated encoder. The adversarial gradients $\nabla_{adv} D$ apparently increase the $V_D$; however, what we want to do is decrease the possibility for correctly classifying the data of encoder with any possible classifier $D_e \in \mathcal{H}$ where $\mathcal{H}$ is hypothesis set of the discriminator. Formally, the true objective is

$$\min \sup_{D_e \in \mathcal{H}} [\log q_{D_e}(s|E(x))]. \tag{2}$$

Alternatively, the problem could be rewritten by the function of the adversarial gradients. Suppose $D^*$ is the best possible classifier given the encoder $E$, and $D'^*$ is the best possible classifier given the encoder $E'$. Then adversarial gradients need to maximize the difference between the $\log q_{D^*}(s|E(x))$ and $\log q_{D'^*}$:

$$V^*(\nabla_{adv} D) = \log q_{D^*}(s|E(x)) - \log q_{D'^*}(s|E'(x)). \tag{3}$$

Then, the following theorem holds.

**Theorem 1.** *The mean square error between $V^*$ and $V_D$ is zero if the expected bias and standard deviation of $q_D$ are equal between the samples of $E'(x)$ and $E(x)$.*

The proof is followed in the appendix. It is worth mentioning that the theorem implies that neither the bias nor the variance of the discriminator over $z \sim E(X)$ is not the deterministic factor of the generalization error. Instead, the generalization error is determined by the difference of the statistical property of the discriminator at two different data points.

## 3 EXPERIMENT

**Datasets:** Following the previous work Iwasawa et al. (2017), the empirical validation uses three user-anonymization tasks on the data of wearables, OppG, OppL (Sagha et al., 2011) and USC (Zhang & Sawchuk, 2012). In all tasks, the neural networks require to learn representations that help activity classification and at the same time, prevent to access the information about users (userID).

**Network Architecture and Training Procedure:** We parameterized the encoder $E$ by convolutional neural networks (CNN) with three convolution-ReLU-pooling repeats followed by one fully connected layer and $M$ by logistic regression, following a previous study Yang et al. (2015); Iwasawa et al. (2017). For all experiments, we apply the post-processing procedures for censoring representation. Specifically, we first trained the encoder $E$ and classifier $M$ with Adam algorithm (Kingma & Ba, 2015) whose learning rate is 0.0001 (150 epochs). Subsequentially, we proceed to the alternative training of $D$ and $E+M$. For the censoring phase, we used RMSprop and we pre-trained the discriminator $D$ for five epochs.

**Baselines:** (1) None: w/o adversary (correspond to standard CNN), (2) Adv: w/ a single adversary, (3) MA: w/ multiple adversaries where each adversary tries to predict from the entire space, and (4)

Table 1: Performance comparison against various $f_{eva}$. Parentheses of value means that the method violates the constrains about accuracy on $Y$. The best performance is shown in bold and underlined.

| | OppG | | | | OppL | | | | USC | | | | |
|---|---|---|---|---|---|---|---|---|---|---|---|---|---|
| $D$ | LR | MLP$_1$ | MLP$_2$ | DNN | LR | MLP$_1$ | MLP$_2$ | DNN | LR | MLP$_1$ | MLP$_2$ | DNN | Avg |
| None | 0.889 | 0.973 | 0.991 | 0.989 | 0.901 | 0.969 | 0.987 | 0.988 | 0.679 | 0.789 | 0.859 | 0.854 | 0.906 |
| Adv-MLP | 0.529 | 0.801 | **0.927** | 0.929 | 0.476 | 0.617 | 0.352 | 0.353 | 0.647 | 0.784 | 0.847 | **0.829** | 0.674 |
| Adv | (0.546) | (0.781) | (0.902) | (0.918) | 0.353 | 0.352 | **0.294** | **0.352** | 0.647 | 0.778 | 0.846 | 0.846 | (0.634)* |
| MA-MLP | 0.567 | 0.805 | 0.945 | 0.938 | 0.353 | 0.353 | 0.792 | 0.353 | 0.624 | 0.775 | 0.846 | 0.843 | 0.683 |
| MA | 0.547 | 0.809 | 0.941 | 0.936 | **0.294** | **0.294** | 0.583 | 0.353 | 0.623 | **0.770** | 0.847 | 0.839 | 0.653 |
| MARS-MLP | **0.486** | **0.786** | 0.945 | **0.910** | 0.352 | 0.353 | 0.353 | **0.294** | **0.620** | 0.771 | **0.841** | 0.836 | 0.629 |
| MARS | 0.476 | **0.720** | **0.915** | **0.904** | **0.294** | **0.294** | **0.294** | 0.353 | 0.622 | **0.766** | 0.844 | 0.824 | **0.609** |

(a) OppG (Balance)  (b) USC (Balance)  (c) USC (Effect)

Figure 1: (a, b)Performance comparison on balance between user-classification accuracy and label-classification accuracy between Adv and MARS with different terminated epoch and $\lambda$. (c)Generalization error of the effect of the adversarial gradients.

MARS: w/ multiple adversaries where each adversary tries to predict from a different subspace of the whole space. Each adversary is parametrized by deep neural networks (DNN) with 400-200 hidden units. If we need to express the type of adversary, we denote it by a suffix [1]. Without mentioning otherwise, we optimized the $\lambda$ from $\{0.05, 0.10, 0.20, 0.50, 1.0\}$ for each baseline to make the fair comparison. Individually, we select the $\lambda$ that did *not* significantly harm $Y$-accuracy (specifically, 2% or less degradation against None) and maximize the level of anonymization.

**Results:** Table 1 list the user classification accuracy for the three tasks with different adversarial classifiers $D$ and evaluator $f_{eva}$. The MLP denote the multi-layer perceptron with 400 hidden units. For evaluator $f_{eva}$, we tested LR, multi-layer perceptron with 50 hidden units (MLP$_1$), multi-layer perceptron with 800 hidden units (MLP$_2$), and deep neural networks with 400-200 hidden units. If some method violates the constraints about accuracy on $Y$ with all $\lambda$, we report the performance with $\lambda$ with gives minimum performance degradation on $Y$, and indicate it by parentheses of value. We can make the following observations. The result shows that MARS or MARS-MLP gives best or second best performance in almost all conditions (pairs of $f_{eva}$ and datasets). It is worth mentioning that Adv, OppG never meat the constraints about Y-accuracy with any $\lambda$.

Figure 1-a, b compares the performance of Adv and MARS with varied $\lambda$ and the number of training epochs (each data point represents different $\lambda$ or the number of iteration). The horizontal axis corresponds to the label-accuracy while vertical axis corresponds to the user accuracy that is evaluated by DNN with five epochs for training. These results show that incorporating MARS is a better strategy for improving the performance of AFL compared with merely tuning $\lambda$ or the number of training epochs. Figure 1-c shows the generalization error of the effect of adversarial gradients. The first row of the figure shows the classification accuracy of the discriminator pre-update of encoder E. The last two row represents the effect of the update E to the accuracy of the discriminator $D$ and the external evaluator $D_e$. The results show that there is enormous overfitting regarding the effect of the adversarial gradients to the $D$ and $D_e$, although this is somewhat alleviated by the proposed method. Most notably, the update to deceive $D$ gives negative effect at the beginning of the training. The correlation between second and third rows is less than 0.06 for Adv with any $\lambda$, and 0.24 for MARS. The result also implies that the accuracy of the discriminator is not the deterministic factor of the success of the AFL since all method gives similar accuracy before the update of the encoder.

## 4 CONCLUSION

This paper proposes an ensemble approach for improving the performance of AFL in the context of the censoring representations. The experimental results show the proposed method gives superior performance. The empirical validations show that the AFL suffered from huge overfitting regarding the effect of the adversarial gradients, suggesting the future assessment is required.

---

[1]For example, Adv-LR means logistic regression parametrizes an adversary.

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

## A  PROOF OF THE THEOREM

*Proof.* Suppose $z'$ is the sample come from $E'(x)$, an $z$ is the sample come from $E(x)$. With standard bias-variance decomposition theorem, expected mean square error of $|V_D - V^*|$ is equivalent to

$$\mathbb{E}[\{|V_D - V^*|\}^2] = \{\mathbb{E}_{\mathcal{D}}[V_D] - V^*\}^2 + \mathbb{E}_{\mathcal{D}}[\{V_D - \mathbb{E}[V_D]\}^2]. \qquad (4)$$

From the definition of the $V_D$ and $V^*$, the first term of eq.4 is equal to

$$\{\mathbb{E}_{\mathcal{D}}[\log q_D(s|z) - \log q_D(s|z')] - \log q_{D^*}(s|z) + \log q_{D'^*}(s|z')\}^2$$
$$= \{bias(z) - bias(z')\}^2,$$

where $bias(z) = \mathbb{E}_{\mathcal{D}}[\log q_D(s|z)] - \log q_{D^*}(s|z)$. The equation is equals to zero if and only if $bias(z) = bias(z')$.

Similar development of the second term of eq.4 proof that the second term equals to zero if and only if the expected standard deviation is equal between $z$ and $z'$. $\qquad \square$

