# OpenReview forum: "Censoring Representations with Multiple-Adversaries over Random Subspaces"
_ICLR.cc/2018/Workshop — Accept_

### Official Review · AnonReviewer2 · 2018-03-01
**Interesting idea**

**Rating:** 6
**Confidence:** 2

**Review:**

This paper studies a new adversarial feature learning method, with an emphasis on formulating the discriminator in a novel manner. It proposes to use an ensemble of discriminators, each of which only uses a subset of the coordinates the encoder outputs. Experiments show that the new discriminator better protects the sensitive features of the data, compared to previous works.

quality: the experimental results seems impressive. I have a few questions:
1. in the experiments , we fix the accuracy degradation rate (2%) and minimize the accuracy of evaluator. Can we give a full tradeoff on these two quantities? This will correspond to applications where different anonymity level is required.
2. Comparing MA with MARS, we see that MARS is better - are there any explanations? Is it because that the models trained using MA are not diverse enough?

clairity: I don't quite follow Section 2.3, especially the notion of adversarial gradients. What is a formal definition of function V_D(.)? How does Equation (2) relate to the equation in the first page? Also, in the proof of Theorem 1, the analysis of the second term, could you elaborate on why the two variances being equal imply that the second term is zero?

originality: I am not familiar with relevant literature, and thus unable to judge.

significance: I think this is a nice step towards understanding the role of discriminator in the study of adversarial feature learning.

minor commments: please correct the author list in the pdf file, and correct the typographical errors.

---

### Official Review · AnonReviewer3 · 2018-03-11
**Nice paper**

**Rating:** 7
**Confidence:** 3

**Review:**

This paper proposed an adversary model, called MARS (multiple-adversaries over random subspace), for adversarial feature learning. MARS consists of several adversaries each in charge of a randomly selected subset of features to make a stronger adversary model. The authors provided theoretical results on the generalization error and empirical results on how MARS outperforms other adversary models by better protecting the sensitive features without losing classification accuracy.

The idea of MARS is quite interesting and the experimental results are nice. It would be better if the paper provides more intuition and explanation of why MARS performs better than those baseline adversaries, and if, for example, there is a connection between the MARS and general ensemble learning, it might be better if more references are added.

The writing is generally clear, though it can be better if all concepts are explained in more detail if space permits.

---

### Official Review · AnonReviewer1 · 2018-03-12
**Modifying the learning objective as in adversarial feature learning works better when many random adversaries are instead instantiated.**

**Rating:** 6
**Confidence:** 3

**Review:**

This is an interesting paper on a simple modification to adversarial feature learning and its consequences. The selection is somewhat of a complementary version of random forests in that subspaces are being chosen to learn from poorly (censor) rather than well.

Theorem 1 may be worth leaving in, but its discussion is not insightful ("It is worth mentioning...") - no statistics of the generated variables with respect to the prior can ever predict generalization independent of the learning algorithm (as is being stated), because both P_g and z are chosen in a highly under-constrained manner by the algorithm.

Please label figures e.g. Fig. 1c does not have column labels (datasets?).

I am not well-versed in recent work on AFL and so cannot easily assess novelty, but if this is novel it is worth knowing. The evaluation follows up on the authors' recent work and is on a very relevant application.

---

### Decision · Program_Chairs · 2018-03-20
**ICLR 2018 Workshop Acceptance Decision**

**Decision:**

Accept

**Comment:**

Congratulations, your paper was accepted to the ICLR workshop.